# EXECUTION-GUIDED WITHIN-PROMPT SEARCH FOR PROGRAMMING-BY-EXAMPLE

**Gust Verbruggen**
Microsoft
Keerbergen, Belgium
gverbruggen@microsoft.com

**Ashish Tiwari, Mukul Singh, Vu Le & Sumit Gulwani**
Microsoft
Redmond, USA
{ashish.tiwari,singhmukul,levu,sumitg}@microsoft.com

## ABSTRACT

Large language models (LLMs) can generate code from examples without being limited to a DSL, but they lack search, as sampled programs are independent. In this paper, we use an LLM as a policy that generates lines of code and then join these lines of code to let the LLM implicitly estimate the value of each of these lines in its next iteration. We further guide the policy and value estimation by executing each line and annotating it with its results on the given examples. This allows us to search for programs within a single (expanding) prompt until a sound program is found, by letting the policy reason in both the syntactic (code) and semantic (execution) space. We evaluate within-prompt search on straight-line Python code generation using five benchmarks across different domains (strings, lists, and arbitrary Python programming problems). We show that the model uses the execution results to guide the search and that within-prompt search performs well at low token budgets. We also analyze how the model behaves as a policy and value, show that it can parallelize the search, and that it can implicitly backtrack over earlier generations.

## 1 INTRODUCTION

Automatically synthesizing code from any form of intent or specification is considered as a holy grail of computer science (Gulwani et al., 2017). One particularly challenging specification are a set of inputs and their expected outputs, as the synthesizer has to not only detect a pattern between the input and the output, but also write code for it. Large language models (LLMs) (Brown et al., 2020; OpenAI, 2024) are trained to recognize patterns in text to predict the most likely next token. Their training data contains significant amounts of code (Xu et al., 2022) and examples of how to use that code, both through tutorials and unit tests.

Consider, for example, a prompt that consists of two `assert` statements and a function signature:

```
assert f(a = 'Program-Synthesis') == 'PS'
assert f(a = 'Large-Language-Model') == 'LLM'

def f(a):
```

Different versions of the `gpt` model series by OpenAI (`4o`, `4-turbo`, `35-turbo`) complete it with a variation on `return ''.join(i[0] for i in a.split('-'))` to make `f` satisfy the assertions.[1]

When the patterns become more complicated, however, the LLM often fails. A first reason is the model overfitting on a dominant (more common) pattern in the assertions. If we change the second input to `'Large--Language--Models'` we notice that `gpt-4o` does not change its answer, not realizing that splitting on `'-'` in a string with `'--'` results in empty strings. The model then only considers the syntactic space, rather than reasoning about the semantics of the program. A second reason is that iteratively sampling tokens does not allow the model to reconsider earlier code like a human would do. Once the model has written `return ''.join(i[0] for i in a.split())` —even if it would now

---

[1]At temperature 0, which we use for all demonstrations.

realize that `i` can be empty—it cannot easily[2] recover. The model is thus not able to *search* for a correct program, and search is indispensable in symbolic synthesizers (Gulwani et al., 2017).

We tackle the limitation on search by sampling multiple lines of code, combining them into a single program, and implicitly letting the model choose which of these lines to continue from in a next iteration. We call this *within-prompt search* as one prompt contains all states explored so far. We tackle the limitation on reasoning about semantics by executing generated code and providing these executions as comments to the model. The model then has access to both the syntax and semantics of the program, and the within-prompt search is thus *execution-guided*.

Considering the previous example and appropriately prompting the model to generate options for the next line of code, the model generates three unique statements:

```
v1 = a.split('-')        v1 = a.split('-')[0][0]        v1 = a[0]
```

These statements are de-duplicated and executed to yield a new prompt for the next iteration:

```
assert f(a = 'Program-Synthesis') == 'PS'   # e1
assert f(a = 'Large--Language--Model') == 'LLM'  # e2

def f(a):
    v1 = a.split('-')  # {'e1': ['Program', 'Synthesis'],
                       #  'e2': ['Large', '', 'Language', '', 'Model']}
    v2 = a[0]  # {'e1': 'P', 'e2': 'L'}
    v
```

The model now completes it with `v3 = ''.join((word[0] for word in v1 if word))`.

Relating this to existing neural program synthesizers, we observe that the model acts as both a policy—by generating candidate lines—and a value—by choosing which previous lines to consider the result of—by looking at both syntax and semantics of generated code. Core differences are that it is not limited to a domain-specific language or even a specific domain—the same method can be used on string transformations (Gulwani, 2011) and lists (Rule et al., 2024) without any intervention—and does not require any training.

We evaluate the effect of within-prompt search and execution-guidance on straight-line program synthesis on five benchmarks spanning three domains (string transformations, list functions, and generic Python programming problems). We provide insights in how the model performs as a policy and value, evaluate the effect of more diversity in sampling lines of code, and compare with a baseline of unrestricted synthesis.

In summary, we make the following contributions:

- We propose *execution-guided within-prompt search* for programming-by-example, which allows the model to reason about the semantics of lines of code while exploring different candidate programs.
- We introduce the *adapted pass@k rate* to evaluate our approach and different baselines on five datasets across various domains.
- We analyze properties of the model as a policy and value, show the value of execution, show that out-of-prompt search (which corresponds to tree-of-thought) scales better with more operations being sampled, show that the model can parallelize the search, and show that it can backtrack to earlier programs.

## 2 BACKGROUND

Given $n$ input-output examples $E = \{(\mathbf{x}_i, y_i)\}_{i=1}^{n}$ the goal of programming-by-example (PBE) is to find a program $P$ such that $P(\mathbf{x}_i) = y_i$ for all $i$. Typically, only a small subset of examples $E_g \subset E$ is given. We say that $P$ is sound if $P(\mathbf{x}_i) = y_i$ for all $(\mathbf{x}_i, y_i) \in E_g$ and we write $P \models E_g$.

---

[2]It could still add a `if '--'not in a else ...` to the end of the line, but we never witnessed this.

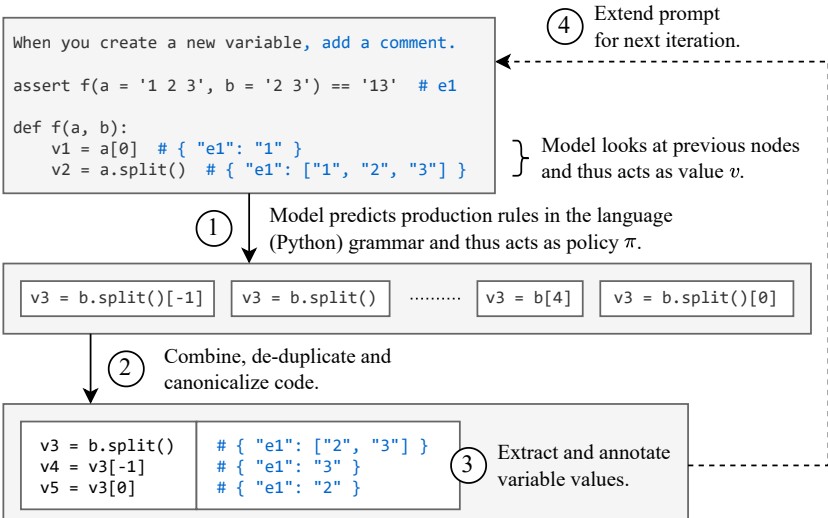

Figure 1: Overview of our approach for PBE with large language models. ① The model acts as a policy by predicting next lines of code. ② We combine, de-duplicate and canonicalize that code in order for the model to act as a value predictor. ③ We execute code, extract values and add them as comments. ④ We extend the prompt with the newly generated code.

**PBE with search** Constraining $P$ to a domain-specific language (DSL) $\mathcal{L}$ defined by a context-free grammar allows us to easily—and sometimes even efficiently—search for programs. FlashFill, likely the most popular application of PBE, traverses its grammar top-down by breaking down the original problem into smaller sub-problems and recursively solving them (Gulwani, 2011). Recent neural approaches perform a bottom-up search by using a policy network to predict one or more production rules to follow, which either initialize a new sub-program (terminal) or combine two subprograms (non-terminal) (Parisotto et al., 2017; Balog et al., 2017; Ellis et al., 2018; 2021; Odena et al., 2021; Chen et al., 2018). Optionally, a separate value network can be used to prioritize or filter candidates in subsequent iterations Ellis et al. (2019).

**PBE with execution** Since each node in a bottom-up search is a program without free variables, it can be executed, and those execution results are a powerful signal for predictions in subsequent search steps. One approach is to add the current execution state as an additional input to the model (Ellis et al., 2019). Another approach is to use the execution of a candidate program $P'(\mathbf{x}_i)$ to define a new problem specification $\{(P'(\mathbf{x}_i), y_i) \mid (\mathbf{x}_i, y_i) \in E_g\}$ for the next iteration (Chen et al., 2018).

**PBE with large language models** Instead of searching over a grammar, some neural approaches generate programs by having the model directly predict tokens from the alphabet of the DSL (Devlin et al., 2017; Bunel et al., 2018). Pre-trained language models also auto-regressively generate tokens and typically see a lot of code during training (OpenAI, 2024). They can be prompted or fine-tuned (Li & Ellis, 2024) to generate programs by example, without being limited to a DSL.

## 3 EXECUTION-GUIDED WITHIN-PROMPT SEARCH

We propose to use large language models as both the (explicit) policy and (implicit) value network to search for code in a bottom-up way, and inject the execution results of that code to guide the model towards better predictions. An overview of this approach is shown in Figure 1. The following sections describe (1) how the model acts as a **policy**, (2) how to prepare that code for the model to implicitly consider the **value** of each line, (3) how to **guide** these results by executing the generated code, and (4) how this constitutes an in-prompt **search**.

## 3.1 POLICY: PREDICTING LINES OF CODE

Consider a function $P(\mathbf{x}) = [o_1, \ldots, o_n]$ where each operation $o_i(o_{<i}, \mathbf{x})$ is a function of the outputs of the previous operations $o_{<i}$ and the input $\mathbf{x}$ that returns a single value (denoted as $o_i$). The value of $o_n$ is the return value. An operation can be drawn from the grammar of any programming language.

Encoding the specification given by $E_g$ and a partial program $P_j = [o_1, \ldots, o_j]$ in a prompt, we can sample a new operation $o_{j+1} \sim p_{\text{LLM}}(\cdot \mid E_g, P_j, t)$ from the LLM with $t$ the sampling temperature (Ackley et al., 1985). If the temperature is set to 0, the model acts as a deterministic policy that maps each state $(E_g, P_j)$ to its most likely next operation $o_{j+1}$. For higher temperatures, it acts as a stochastic policy that maps $(E_g, P_j)$ to a distribution of operations.

Instead of a learned policy $\pi$, the model is prompted to find $\arg\max_{o_j \sim \pi} [Q^\pi(o_j, P_{j-1})]$ where the value $Q^\pi(o_j, P_{j-1})$ of considering an operation $o_j$ in program $P_{j-1}$ is the expectation that it can be completed into a valid program, or $Q(o_j, P_{j-1}) = \mathbb{E}_{P_\tau \sim \pi} [P_{j-1} \circ o_j \circ P_\tau \models E_g]$ where $P_\tau$ is a sequence of operations sampled from the (prompted) policy.

**Example 1** *The specification $E_g$ and program $P_2$ are encoded in the prompt on the right. An LLM will auto-complete this program with new statements*

$o_3^1 = $ `b.split()[-1]`     $o_3^2 = $ `b.split()`

$o_3^3 = $ `b[4]`     $o_3^4 = $ `b.split()[0]`

*that correspond to operations.*

```
assert f(a = '1 2 3', b = '2 3') == '13'

def f(a, b):                          E_g
    v1 = a[0]
    v2 = a.split()  } P_2
    v3 =
```

## 3.2 VALUE: COMBINING LINES OF CODE

Suppose we sample $k$ operations $O_{j+1} = \{o_{j+1}^1, \ldots, o_{j+1}^k\}$ from the language model with $t > 0$. Instead of heuristically picking one operation $o_{j+1}^* \in O_{j+1}$ to extend $P_j$ before sampling the next operation—writing such heuristic is hard even without considering the different types of values that each operation can return—we set $P_{j+k'} = f_C(P_j \circ O_{j+1})$ with $k' \leq k$ and let the model act as that heuristic in the next iteration (where $\circ$ denotes operation concatenation). Here, $f_C$ is a function that de-duplicates and canonicalizes the given operations, resulting in a potential subset $k'$ of the $k$ new operations to be added.

Instead of a single program, $P_j$ then represents all programs explored in all iterations $\leq j$, where each operation can be considered as its own root node. When writing an operation $o_{j+1}$, the model needs to select some nodes $o_i \in P_j$ to continue from. Ideally, it selects operations that are expected to lead to a correct program by implicitly considering $V^\pi(o_i) = \mathbb{E}_{o \sim \pi} [Q^\pi(o, P_i)]$ to estimate the expected value of each operation $o_i \in P_j$. We call this *within-prompt search* (WPS) as the search happens within one prompt that is iteratively extended.

**Example 2** *Continuing the example, the implicit value function assigns a high score to* `b`, *as it is used in all operations. Since $o_3^3$ does not execute, it is removed. Operation $o_3^2$ is a subprogram of $o_3^1$ and $o_3^4$ and the substitution $o_3^1 = o_3^2$.`split()[-1]` and $o_3^4 = o_3^2$.`split()[0]` is made by $f_C$. $P_5$ is shown on the right. The value function can decide which state to continue from in the next iteration.*

```
def f(a, b):
    v1 = a[0]
    v2 = a.split()
    v3 = b.split()
    v4 = v3[-1]
    v5 = v3[0]
```

We can also use the model as an explicit value function by using a separate prompt to rank each $o_{j+1}^i \in O_{j+1}$ and getting $k$ next states $P_{j+1}^i = P_j \circ o_{j+1}^i$ that we can explicitly search over based on their value. This corresponds to Tree-of-Thought (Yao et al., 2024) with lines of code corresponding to *thought decompositions*, the policy corresponding to the *thought generator* and the value prompt being the *state evaluator*.

## 3.3 GUIDANCE: EXECUTING CODE

Understanding the execution semantics of $P_j$ is crucial when deciding on a next operation $o_{j+1}$—especially since the specification $E_g$ is in this semantic space. Previous work (Ellis et al., 2019) has shown that executing code $[\![P_j]\!] = \{P(\mathbf{x}) \mid \mathbf{x} \in E_g\}$ on the given examples $E_g$ and conditioning the policy and value on $([\![P_j]\!], E_g)$ instead of $(P_j, E_g)$ allows them to make better decisions in

this semantic (*execution*) space. The intuition is to mimic a human developer using code printing statements to evaluate their progress so far, and then make decisions based on that progress.

We extend this idea to our prompted policy and, as LLMs have shown strong code understanding capabilities, additionally allow the model to make decisions based on both the semantic and syntactic representation of $P_j$. Instead of only the final output $[\![P_j]\!]$, we obtain the value of each operation $o \in P_j$ on the given examples and add $[\![O_j]\!] = \{\{o(\mathbf{x}) \mid \mathbf{x} \in E_g\} \mid o \in O_j\}$ to the condition of our policy—to the prompt.

**Example 3** *Before sampling new operations, we execute $P_5$ and annotate each line (operation) with its output on the given input (e1) to allow reasoning about both the syntax (code) and semantics (output) of the current states.*

```
def f(a, b):
    v1 = a[0]      # { "e1": "1" }
    v2 = a.split() # { "e1": ["1", "2", "3"] }
    v3 = b.split() # { "e1": ["2", "3"] }
    v4 = v3[-1]    # { "e1": "3" }
    v5 = v3[0]     # { "e1": "2" }
```

### 3.4 SEARCH

Sampling multiple operations $O_j$ and combining them into a new state $(E_g, P_{j+1}, [\![O_{j+1}]\!])$ allows us to perform a within-prompt search. Each state (program) in the search is combined into a single *super*state, which the model can expand. We stop the search when $o_l \models E_g$ for any $o_l \in P_j$ and remove all operations that are not used by $o_l$. We can keep expanding the state until the context window of the LLM is reached and rely on *implicit* backtracking where the (implicit) value function simply ignores operations that cannot lead to a valid program.

**Example 4** *Instead of keeping track of different states, the "search" consists of a single prompt (*super*state) that is iteratively expanded (for l iterations).*

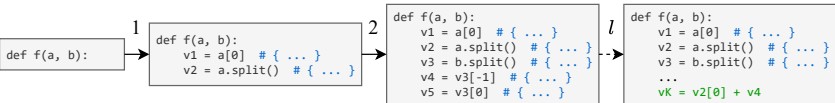

## 4 EXPERIMENTS

We evaluate the effect of execution and search, show insights in how the model behaves as a policy and value, and describe other properties of the search.

### 4.1 IMPLEMENTATION DETAILS

We use Python as the programming language in which we synthesize programs.

**Prompt**  The prompt consists of four parts: (1) an instruction, (2) a single static example, (3) a set of assertions on a function f, and (4) the function f itself. The instruction instructs to generate code without control flow statements, that any function from the standard library can be used, that it can use variables as a scratchpad, and to create a comment each time a new variable is initialized. The example contains one line that is not needed. We initialize each line with " v" to ensure that the model generates a straight-line program with variables (v1, v2, ...) as the model tokenizes digits separately—v1 is always broken down into v and 1. A full example is shown in the Appendix.[3]

**Sampling operations**  As the model is instructed to generate each statement on a single line, we can sample a single operation by using the newline character "\n" as a stopping token. Operations that do not parse or execute are skipped.

**Canonicalization**  After ensuring that each generated line of code assigns its value to a unique variable, we concatenate these lines and perform global canonicalization steps. All comparisons are made on the abstract syntax tree (AST) level.

1. For every assignment v = r, we replace any occurrence of r in another operation with v and ensure that the lines remain topologically sorted.

---

[3]See the supplementary material.

2. For every assignment `v = r` where `r` is a `generator`, which does not have a pretty printing implementation due to lazy execution semantics, we wrap it in a `list` call as `v = list(r)`.

3. We remove any duplicate lines of code based on their execution results.

**Execution and comments**  We assign assertion `assert f(x1) == y1` a unique identifier `e1` as a comment "`# e1`". This allows us to easily anchor the execution result of each operation `v = r(x)` on to that identifier in a dictionary by mapping `{ "e1": r(x1) }`. Ensuring that each value can be printed is done in the canonicalization step to not have a disconnect between the operation and the commented values.

**Search**  The search stops when any of the variable results matches the assertions or when a limit on the number of iterations is reached.

## 4.2 EXPERIMENTAL SETUP

**Datasets**  We use five popular datasets that span different domains. No changes are required to apply our approach to these different datasets.

- **PROSE** (Microsoft) is a set of 354 benchmarks originally used to evaluate FlashFill Gulwani (2011)—the first widely adapted application of PBE. We filter it down to 332 non-trivial problems (no empty values, three or more examples).

- **SyGuS** (Alur et al., 2019) is a set of 205 benchmarks from the string transformation track of the SyGuS Alur et al. (2013) competition.

- **Playgol** (Cropper, 2019) is a set of 327 benchmarks from inductive logic programming.

- **Lists** (Rule et al., 2024) is a dataset of 251 list understanding problems, where each task transforms a list of integers.

- **MBPP** (Austin et al., 2021) is a dataset of basic Python problems, where each problem consists of both a description in natural language and a set of assertions. These assertions are given as part of the input, mostly to provide any synthesizer with information about the expected signature of the function to be synthesized. We only use the assertions, remove any that are not simply evaluating `f(x) == y` with $x \in \mathbf{x}$ and y constants that can be evaluated with `eval` without additional imports and that are `json` serializable (no `set`, no `datetime`). Across the train and test set, this leaves us with 382 problems.

To save on cost of prompting for experiments, we filter *trivial* benchmarks by using a simple prompt conditioned on the first example $(\mathbf{x}_1, y_1) \in E$ to sample five programs $P^i$ at $t = 0.6$ and do not include a benchmark if $P^i \models E$ for all of them. This leaves 238 benchmarks from PROSE, 94 benchmarks for SyGus, 170 benchmarks for Playgol, 211 for Lists, and 268 for MBPP.

**Hyperparameters**  Unless specified otherwise, we sample $k = 4$ operations at each line at a temperature of 0.6, which is a common value that achieves a nice balance between exploration and exploitation. The model is `gpt-4o`. We set the maximal number of iterations to 8.

**Baselines and variations**  We consider four baselines on the task of straight-line code generation.

- **Straight** directly asks the model to generate the code.

- **Chain-of-thought (CoT)** (Wei et al., 2022) asks the model to first explain its thoughts and then generate code. This does not use any execution feedback or search.

- **Self-debug** (Chen et al., 2023; Wang et al., 2024) executes the code on the examples and iteratively asks the model to refine based on the outcome. When an exception is thrown, only exceptions are shown. When no exception is thrown, all wrong outputs are reported as `f(x) = y'` (expected y). We only show the last iteration to the model, so any prompt in iteration $i > 1$ is structured as [system, user$_1$, assistant$_{i-1}$, user$_{i-1}$] with assistant$_{i-1}$ program after $i-1$ iterations and user$_{i-1}$ the feedback on that program. We use 8 iterations.

- **Tree-of-Thought (ToT)** (Yao et al., 2024) uses an explicit value prompt to perform a search. As opposed to our within-prompt search, this perform an outside-prompt search.

The value prompt asks the model to reason about each line (including its execution) and rate it with *sure*, *maybe* or *impossible* (like the original ToT). Nodes are scored by value of their last line first and depth second, meaning that the search backjumps to the last unexplored node in which the last line has the highest score.

**Metrics** The *pass@k* rate——the probability that at least one out of $k$ samples is correct— is a common metric in code generation (Chen et al., 2021a). Because each method implicitly generates a different number of programs to arrive at one completion, we compute the *aligned pass@8* rate. For example, straight only generates one program per completion, but within-prompt search generates 4 programs with $k = 4$, so we compare their pass@4 and pass@1 rates, respectively. This corresponds to using a generate-and-test strategy for cheaper methods until the same budget is reached. We set the budget to 8 and align the metrics as follows. **Straight:** pass@8 over 32 completions. **CoT:** pass@4 (the thought counts as one program, despite consuming significantly more tokens) over 16 completions. **Self-debug:** pass@1 repeated over 4 folds. **ToT:** pass@1 (the value prompt counts as one program as it includes reasoning) repeated over 3 folds. **WPS:** pass@2 repeated over 4 folds.

## 4.3 RESULTS

**Search and execution yields better programs.** Figure 2 shows the aligned pass@4 rate of straight-line code generation with search, execution, and both. Execution without search is done at $k = 1$ with temperature 0 and is thus deterministic (pass@1). Without execution, the implicit value function does not help the model select better completions, and the performance is similar to straight-line code generation. With execution and search, within-prompt search (pass@1) allows the model to perform better than the generate-and-test strategy (pass@4) in 4 out of 5 benchmarks. On the Lists benchmark, where the problem statement is less clear from looking at examples, generating candidates as a single program is more effective than trying to reason about the intermediate values.

**Within-prompt search generates good code on a budget.** Figure 3 shows the aligned pass@8 of all approaches. Within-prompt search is either the best (3/5) or second-best (2/5) generation strategy. SyGus has a set of similar problems with long strings, where different variations of splitting are generated in the first iteration, which results in many small value tokens that confuse WPS. By reasoning about which is the correct split, ToT performs better at these. Conversely, on Lists, it is often less clear what the problem is, and either trying different things (WPS) or getting directed, end-to-end feedback (self-debug) perform better. Thinking about the problem (CoT) or a generate-and-test strategy (straight) are also effective.

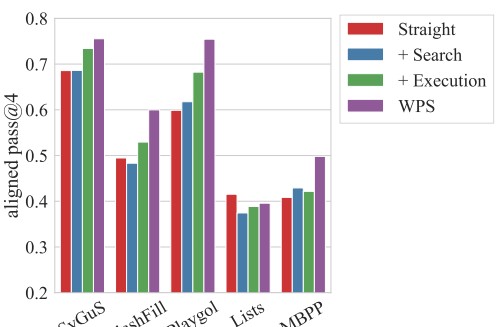
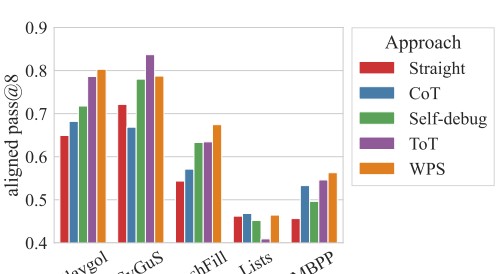

Figure 2: Without execution results, search (pass@1) does not improve over straight-line code generation (pass@4). Execution helps both without search (deterministic pass@1) and especially if the model can perform within-prompt search (pass@1).

Figure 3: Comparing different baselines on straight-line code generation. Within-prompt search (WPS) is consistently among the top-performers, achieving best scores on 3/5 benchmarks.

**Explicit value prompt scales better with number of operations.** Figure 4 shows how the pass@1 changes in function of the number of lines sampled for a diverse—where we sampled lines

with increasing temperature until exactly $k$ syntactically unique lines are obtained—and default sampling strategy. First, we observe that in most settings, performance improves with $k$, showing that the model can serve as both an explicit (ToT) and implicit (WPS) value function. Second, we observe that for the same $k$, the explicit policy (with higher budget) performs better, especially as $k$ increases. The number of choices scales linearly with the iteration $i$, and the number of tokens scales as $i * n$ with $n$ the number of examples, both of which make the problem of selecting good nodes more difficult.

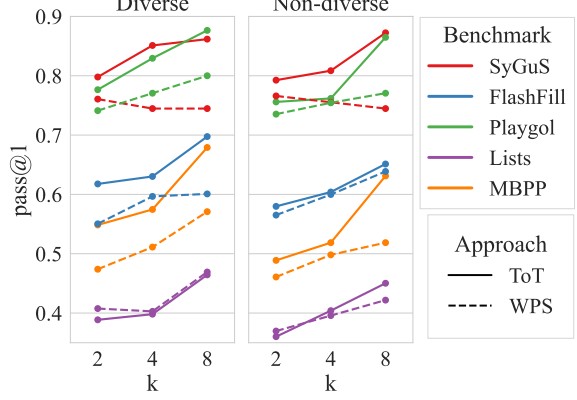

Figure 4: pass@1 in function of $k$ for diverse (left) and non-diverse (right) policy. Explicitly computing the value of new lines (ToT) scales better with increasing the number of operations.

Figure 5: Number of emitted tokens in function of number of iterations for different $k$. The explicit value function (ToT) requires roughly twice the amount of tokens.

**Prompted model behaves as a trained policy**   Figure 6 shows the correctness in function of the average number of statements sampled in the first or any iteration by the policy. There is a clear negative correlation: if the policy is certain and samples with less diversity, the model is more likely to solve a problem in the end. Conversely, if the policy samples many diverse lines, it is less likely to solve the problem in the end. This indicates that the model does behave like a policy that was trained to prefer lines with higher values.

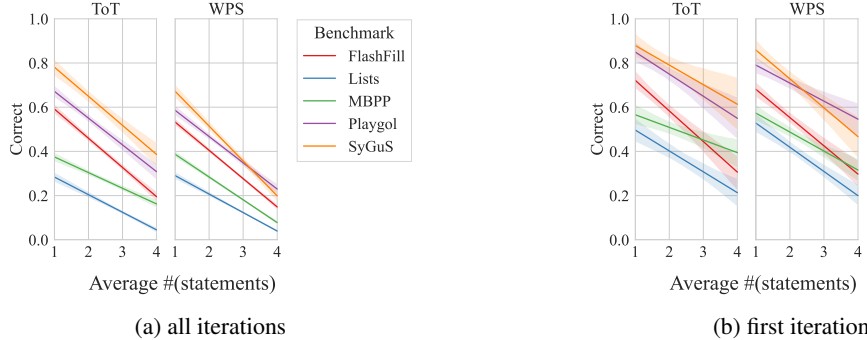

(a) all iterations

(b) first iteration

Figure 6: Regression lines and 95% confidence interval of pass@1 rate in function of the average number of unique operations sampled in (a) any or (b) the first iteration. The first operation (b) is less important, as there is still room to recover.

**Within-prompt search works in parallel**   Figure 7 shows the number of iterations required to find a successful program of a given length. Some of the weight is above the identity line, where the final program uses two or more nodes from the same iteration at least once. The stochastic policy thus allows within-prompt search to consider different leaf nodes in parallel. This is confirmed

by Figure 8, which shows a distribution of the number of iterations taken to solve success cases. Within-prompt search solves more of its cases in fewer iterations.

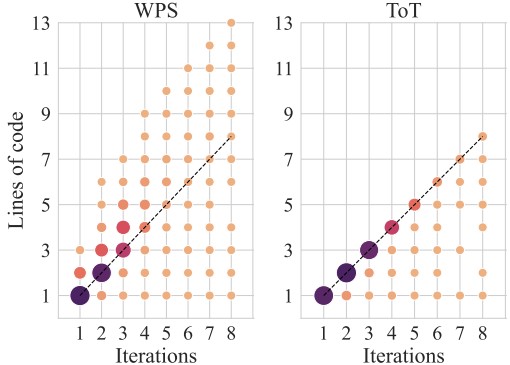
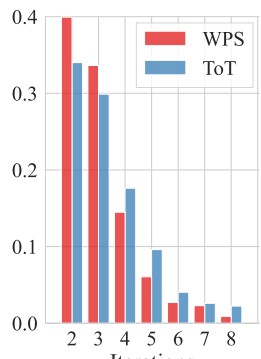

Figure 7: Length of successful programs versus the iteration in which they were found for WPS (left) and ToT (right). Above the line, the policy returned multiple operations that were used in the final program. Below the line, the policy either failed to generate any new results or the value back-tracked and completely ignored the results from one iteration.

Figure 8: Distributions of number of iterations to solve problems for WPS and ToT. Parallelism in within-prompt search allows problems to be solved in fewer iterations.

**Within-prompt search can backtrack**  Cases below the line in Figure 8 have iterations without any effect: either the policy failed to generate a (valid) new line or the value function backtracked and completely ignored the results from one iteration. Figure 9 shows how often ToT explicitly backtracked and how often ToT and WPS implicitly backtracked, where an implicit backtrack is defined as an iteration that did not yield any operations that were kept in the final program.

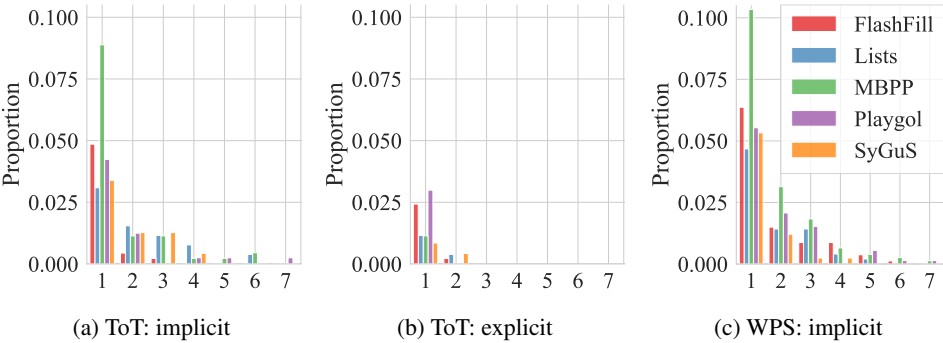

Figure 9: (a) Implicit and (b) explicit backtracking for ToT, and implicit (c) backtracking for WPS in successful solves. In general, backtracking is quite rare, happening in around 10% of problems. It happens slightly more for WPS, where having more nodes to choose from causes more dead ends.

**Example 5** *On the problem of determining whether two lists contain the same number of elements, the model takes three iterations (with many duplicate lines) to figure out that it does simply need to compare the lengths of the inputs. In the fourth iteration, it immediately suggests the correct line, backtracking over iterations (2) and (3).*

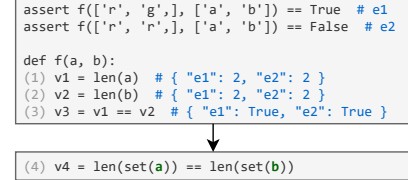

```
assert f(['r', 'g',], ['a', 'b']) == True   # e1
assert f(['r', 'r',], ['a', 'b']) == False  # e2

def f(a, b):
(1) v1 = len(a)    # { "e1": 2, "e2": 2 }
(2) v2 = len(b)    # { "e1": 2, "e2": 2 }
(3) v3 = v1 == v2  # { "e1": True, "e2": True }
```

```
(4) v4 = len(set(a)) == len(set(b))
```

## 5  RELATED WORK

**Programming-by-example with search**  Two popular search-based strategies for programming-by-example approaches are *bottom-up* synthesis and *top-down* synthesis (Gulwani et al., 2017).

In forward synthesis, the search space is traversed by combining subprograms into more complex programs through enumeration (Alur et al., 2017) or enumeration with priority defined by a neural network (Odena et al., 2021; Ellis et al., 2021) or through an incomplete search powered by neural network heuristics (Ellis et al., 2019; Shi et al., 2022; 2023).

**Programming-by-example with language models**  Without search, auto-regressive neural networks can also be trained to output the program token by token (Devlin et al., 2017; Bunel et al., 2018). This is particularly interesting with large, pre-trained language models, which can be fine-tuned on synthetic data (Li & Ellis, 2024). Models that are large enough to exhibit in-context learning abilities, such as the GPT family (Brown et al., 2020; OpenAI, 2024), can program by example through prompts. One can use chain-of-thought prompting too by first asking the model to describe the transformation problem in natural language (Wang et al., 2024).

## 6    LIMITATIONS AND FUTURE WORK

Empirical, small-scale experiments show that older (like `gpt-35-turbo`) and smaller models (like CodeLlama (Roziere et al., 2023)) perform significantly worse, because they either do not adhere to the instruction to generate code line by line or because they are unable to properly reason about code and comments. For this reason, we did not perform experiments on these models. A potential solution is fine-tuning these smaller models as a policy, for example, on straight-line programs obtained by a teacher model on synthetic data (Li & Ellis, 2024).

Our implementation considers straight-line code generation, which provides a controlled environment for evaluating different search strategies. Extending within-prompt search to any code requires streaming access to the language model, which allows dynamically stopping when a full operation is sampled, and special care for control flow statements, where we can take inspiration from previous work on teaching LLMs to reason about code execution for program repair (Ni et al., 2024).

Within-prompt search can be applied to other problems, such as program repair (Ni et al., 2024) or code auto-completion in environments where inputs are available, like spreadsheets (Chen et al., 2021b) or data science notebooks (Huang et al., 2024). As there is no test for soundness, the model then also serves as an implicit reward model when stopping the search.

## 7    CONCLUSION

We introduce execution-guided within-prompt search for programming-by-example with LLMs, where the premise is to sample multiple atomic operations, combine them into a single program, and repeat the process from that combined program. Sampling multiple operations, where the model acts as a policy that explores different next states, allows us to perform a search. Combining operations into a single program enables the search to happen within one expanding prompt. This allows the model to act as a value function that evaluates which of these operations are promising candidates to extend in next iterations. We evaluate this approach on five datasets and different baselines using the aligned pass@$k$ rate, which aligns $k$ according to the token consumption for each method. Our evaluation shows that within-prompt search performs consistently well across benchmarks for the same token budget. If we allow more budget, explicitly computing the value of completions for an out-of-prompt search (which corresponds to tree-of-thought) achieves a better pass@1 rate.

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
