# OpenReview forum: "Execution-guided within-prompt search for programming-by-example"
_ICLR.cc/2025/Conference — ICLR 2025 Poster_

### Official Review · Reviewer_g8hP · 2024-10-31

**Soundness:** 2
**Presentation:** 1
**Contribution:** 2
**Rating:** 5
**Confidence:** 4

**Summary:**

This paper targets a form of program synthesis called programming by example (PBE). The setup includes a specification, given via assert statements that encode the input and their respective outputs. The paper proposes an execution-guided technique to guide the synthesis of the program using LLMs by executing each statement of the generated code and providing the result in the prompt itself to inform the generation of the next statement. The paper compares this technique with standard prompting techniques and show that the execution and search strategy improves program accuracy using GPT-4o as the base LLM.

**Strengths:**

In general, I really like the idea of enabling PBE-style code synthesis with LLMs, especially the implications if we combine PBE with typical NL specifications. I also think the overarching contribution of specifying individual execution results within the prompt is good, and it allows for more fine-grained feedback while generating lines of code.

**Weaknesses:**

I feel that while the core ideas proposed are neat, the paper lacks clarity and depth in terms of explaining the technique proposed and evaluating it. I'll describe my major complaints below.

## Technique

This section does not do a good job explaining the proposed technique clearly. The text is dense and filled with technical notation, but understanding how they interact and play with each other is not easy. It would have been better to include a diagram of all the phases that contained more details than Figure 1, or even better, an algorithm defining each phase and the steps within the phase along with supporting text.

There are also several poorly defined terms and inconsistencies in notation (defining what a policy, value, program, function, operation, etc. are). Furthermore, the cyclic nature of the technique shown in Figure 1 is not explained in Section 3, so I have no idea how the search space is represented within the prompt, how it grows with each response from the LLM, how it is pruned, to what extent can it be pruned, how backtracking works here (the text doesn't describe backtracking but instead describes operations being pruned).

In summary, section 3 provides some context and details but getting the full picture of the system is very difficult. The paper is significantly under the page limit, so it should be able to clearly detail the entire system without sacrificing the details it already presents.
I have asked specific clarifying questions in the Questions section.

## Evaluation

### Baselines and Benchmarks

The evaluation in this paper is lacking. Even before getting into the details, this paper only evaluates its technique over a single base LLM model. At minimum, I would expect more LLMs of varying techniques and sizes to show up in the evaluation. Also, I would like a more in-depth evaluation of different prompting techniques: standard prompting is just not enough. At minimum, chain-of-thought prompting needs to be there, since it is a pretty standard approach, and I would appreciate other baselines that may directly compare with this paper (some alternate PBE techniques are mentioned in the background section). I do like the diversity of the datasets being evaluated. However, I would like more details about the benchmarks and why they were chosen: what are the challenges that each benchmark provides? How many examples do you need to give, and is it different for each benchmark? what language is each benchmark targeting? etc.

### Metrics

The metrics are not defined properly. First, the pass@k metric has a very specific definition already established in the Codex paper[1]. This metric samples the top k responses from an LLM and measures the probability of at least one sample being correct. That means the metric used in the paper is more like pass@8 (since it is considering 8 responses, but I am not sure if that is necessarily the top-8 responses), or equivalently pass@3. I would like to know the results for different values of k for all baselines with and without search and execution. I also don't think all@3 metric has been defined, nor has the notation being used in 4.3 (line 307/308) been defined.

### Results

I would like a bit more insight drawn from the results. For instance, why does SyGuS have a lower performance for +S+X versus +X? Is there any reason SyGuS and Playgol  have higher accuracies for +X versus +S, while for other datasets it is lower? Also how does standard CoT prompting perform here?

In figure 3, what do the horizontal lines indicate? What exactly does a backtrack mean for Figure 4? What is the difference between WPS and Straight/Free? And in Figure 5, what does it mean by average number of sampled operations? I thought in each case a fixed number of operations were sampled?

## Miscellaneous

In general, there are several grammatical mistakes throughout the paper that make reading it a bit difficult. This is not as major from a technical standpoint, but it is important to correct them.

I also think the background section and related work can be explored more. Especially in related work, explain how your technique contrasts with those existing techniques, and why they are not included in the evaluation. I also think there needs to be a section that talks about LLMs for code generation as well, not just PBE.

**Questions:**

These are mainly questions to clarify issues I have from section 3, but please also respond to the concerns from the Weakness section.

1. What is the concrete definition of a policy and a value? And what is the difference between an actual value and a value network?
2. What does an operation mean as far as actual code is concerned? i.e. is it a token, statement, code snippet, etc?
3. What is the difference between P(x) and P_j? And is P_j a partial program or a representation of the entire explored search tree at this point?
4. At each stage, you generate operation o_j as per 3.1. But you calculate expectation wrt P_T, which "is a sequence of operations sampled from the (prompted) policy". So is P_T actually sampled at each point on top of o_j, or is P_T just a statistical notion of the possible ways to complete the current program P_{j-1} o o_j?
5. What constitutes an iteration? Is it the entire process included in steps 1--4 from Figure 1 or is it the generation of a single operation?
6. How is the search space explored? Is it more of a BFS search or a DFS search? I notice mentions of backtracking in 3.4 so I assume it is a form of DFS, but in this case how does the backtracking work? At each point does the LLM generate just the response to one unexplored node or responses for all unexplored nodes? and what is the size of the search tree in general? I'd assume it blows up in complexity for large target programs which is why there is the pruning used. Furthermore what do you mean by "operations not used" as far as pruning is concerned? In general I think you can do a better job in section 3 by grounding each step with an example, or at least referring back to Figure 1 on top of a general algorithm.
7. Do you have any safeguards against executing problematic/malicious code? How about results of for loops, incomplete branches, and more complex control flows that may not necessarily terminate?

---

> ### Author Response · Authors · 2024-11-23
> **Response to review**
>
> Regarding your comment: "This section does not do a good job explaining the proposed technique clearly. [...]"
> - We will add a leading example to the end of each component that clearly demonstrates how everything works together. Figure 1 contains almost all details, except for the implementation details in Section 4.1 and the fact that step 4 is simply a concatenation of step 3 to the initial prompt. We can extend this figure to include the prompt for the next iteration.
>
> Regarding your comment: "There are also several poorly defined terms and inconsistencies in notation (defining what a policy, value, program, function, operation, etc. are). [...]"
> - We apologize for some inconsistent use of terms like function and program that hindered in understanding our contribution. We will fix such issues. We hope the common response above helps in conveying the main ideas. We are mostly using the standard definitions of policy and value functions reinforcement learning. Program and operation are defined on line 164.
> - We will indeed “close the loop” by adding a full example. The search space is represented exactly like in Figure 1. The prompt for the next iteration is obtained by simply concatenating the new (combined) operations, because all states are represented in one “super-state” of a bloated program.
>
> Regarding your comment: "The evaluation in this paper is lacking. [...] At minimum, chain-of-thought prompting needs to be there, [...]."
> - Note that our claim is not to perform better than any domain-specific approach.  Our claim is that (1) language models perform better when they can reason about the intermediate results, and (2) language models can perform within prompt search. These claims result in a method that can be applied to any domain. We have significantly extended our evaluation with controlled experiments on other search strategies and adapted the “simple” strategies to generate-and-test with an aligned budget. Please see the general comment for more information.
> - Regarding different models, initial exploration shows that the model does not perform well on smaller models, because they do not adhere to the specification, without fine-tuning for the operation completion task. We leave this as future work, and will make this limitation explicit.
>
> Regarding your comment: "I do like the diversity of the datasets being evaluated. However, I would like more details about the benchmarks and why they were chosen: what are the challenges that each benchmark provides? How many examples do you need to give, and is it different for each benchmark? what language is each benchmark targeting? etc. "
> - They were chosen to represent different domains, from common string transformation benchmarks (PROSE, PlayGol and SyGuS) to benchmarks where the task is sometimes very hard to understand (Lists and MBPP). MBPP was specifically added to highlight the DSL-free aspect of synthesis with LLMs. Besides MBPP, none of these benchmarks target a specific language. We targeted Python.
>
> Regarding your comment: "The metrics are not defined properly. First, the pass@k metric has a very specific definition already established in the Codex paper[1]. This metric samples the top k responses from an LLM and measures the probability of at least one sample being correct. That means the metric used in the paper is more like pass@8 (since it is considering 8 responses, but I am not sure if that is necessarily the top-8 responses), or equivalently pass@3. I would like to know the results for different values of k for all baselines with and without search and execution. I also don't think all@3 metric has been defined, nor has the notation being used in 4.3 (line 307/308) been defined."
> - To compute pass@k, we need to sample N > k completions and then compute the probability that any subset of k completions contains at least one correct one. If N = k, as you suggest, this collapses to 0 (none is correct) or 1 (one is correct). We set N = 8 and k = 1 to compute pass@1, estimated over 8 completions (simple baselines) and 3 repetitions (anything with iterations).
> - We have updated the metrics to reflect the budget that each method requires. For example, getting 4 completions at each iteration essentially generates 4 programs in total, so we would compare the pass@1 rate of our method with the pass@4 rate of the “simple” baseline. See the general comment for more information.
> - We note that the notation (n k) is standard notation for the binomial coefficient of “n choose k,” but we have removed the all@3 results.

---

> > ### Author Response · Authors · 2024-11-23
> > **Continuation of the review response**
> >
> > Regarding your comment: "I would like a bit more insight drawn from the results. For instance, why does SyGuS have a lower performance for +S+X versus +X? [...]"
> > - We have added more insights into the results for specific “odd” occurrences.
> >
> > Regarding your comment: "In figure 3, what do the horizontal lines indicate?"
> > - We added some vertical jitter to improve the presentation, but are replacing this figure with a heatmap, as suggested by another reviewer.
> >
> > Regarding your question: "What exactly does a backtrack mean for Figure 4? What is the difference between WPS and Straight/Free?"
> > - We have made our notation in figures more consistent. We have also updated the definition of a backtrack: it is now an iteration that yields zero lines of code used in the final program (implicit backtrack). In tree-of-thought, this can also happen explicitly if all candidates receive a low score, which will cause the model to backjump to an unexplored node where the last line has a high score.
> >
> > Regarding your comment: "And in Figure 5, what does it mean by average number of sampled operations? I thought in each case a fixed number of operations were sampled?"
> > - The average number of unique operations, we will add that explicitly. If we get 4 different operations at each iteration, the model is less certain than if it would have sampled 4 identical operations (policy is so certain that it does not do any search).
> >
> > Regarding your question: "What does an operation mean as far as actual code is concerned? i.e. is it a token, statement, code snippet, etc?"
> > - In theory, an operation is any code that takes some outputs from previous operations and returns a new value. In practice, it is currently limited to a single-line Python expression (the right-hand side of an assignment). Some examples are provided in Figure 1.
> >
> > Regarding your question: "What is the concrete definition of a policy and a value? And what is the difference between an actual value and a value network?"
> > - We consider a stochastic policy, which is a function that takes the state (a current program) and returns an action to take (a next line of code that can be concatenated to the previous state to obtain a new state). We will update our notation to make it more explicit that “executing an action” in the environment is simply adding the new line of code to the previous state. The value of a node is the expected return (finding a correct program or not) after starting in that node and following some policy. The value function computes that value for a given node and policy. A value network is a neural network trained to approximate the value function in deep reinforcement learning.
> >
> > Regarding your question: "What is the difference between P(x) and P_j? And is P_j a partial program or a representation of the entire explored search tree at this point?"
> > - P(x) is the result of executing program P on input x, obtained by taking the value of the last operation. We will clarify this in the text. P_j is a partial program that consists of the first j operations. This partial program is a representation of the search tree at that point, with every line (= operation) an explored node.
> >
> > Regarding your question: "At each stage, you generate operation o_j as per 3.1. [...] is P_T just a statistical notion of the possible ways to complete the current program P_{j-1} o o_j? "
> > - P_T is a rollout: a notion of the possible ways to complete the current program to any extension according to the policy. This is how one computes the value of a node (see question 2). We’re not actually approximating this expectation explicitly; we are stating that the model (hopefully) samples operations according to this expectation: operations that it thinks will become a valid program.
> >
> > Regarding your question: "How is the search space explored? Is it more of a BFS search or a DFS search? [...] "
> > - We will add a complete example to each step, to make the method clearer.
> > - The search space is explored implicitly by the model by evolving a single prompt. As mentioned before, the partial program contains all previously explored states, and the model can choose to continue from any state. The search is likely more of a greedy beam search, with the model using some internal heuristic to choose which node to continue from. Empirical evidence shows that it usually prefers to go depth-first, but theoretically it can perform any combination of BFS and DFS. We are open to performing a deeper analysis.
> > - Backtracking is thus also implicit: if the model decides to ignore some nodes and restart with some initial nodes. For example, if it has already computed 8 lines v1 – v8 and v9 is only a function of v1 and v2, it has “backtracked” over the explorations done in v3 – v8.

---

> > > ### Author Response · Authors · 2024-11-23
> > > **Continuation of review response**
> > >
> > > Regarding your question: "Do you have any safeguards against executing problematic/malicious code? How about results of for loops, incomplete branches, and more complex control flows that may not necessarily terminate?"
> > > - There are no safeguards. Any method that uses execution should be cautious of malicious code. But: limiting the model to straight-line code significantly reduces the opportunity for it to accidentally generate bad code.

---

> > > > ### Comment · Reviewer_g8hP · 2024-11-23
> > > >
> > > > Thanks for the detailed response! Have you uploaded a revised version of the paper already? If so can you upload a version highlighting the actual revisions in the text? It becomes difficult to judge the changes otherwise.
> > > >
> > > > One thing I am still unsure of is the pass@k metric you are using. Maybe I am misunderstanding a few things, but here is the definition from the Codex paper:
> > > >
> > > > ```python
> > > > def pass_at_k(n, c, k):
> > > >   """
> > > >   :param n: total number of samples
> > > >   :param c: number of correct samples
> > > >   :param k: k in pass@$k$
> > > >   """
> > > >   if n - c < k: return 1.0
> > > >   return 1.0 - np.prod(1.0 - k /
> > > >     np.arange(n - c + 1, n + 1))
> > > > ```
> > > >
> > > > Can you explain how this maps to your system? Basically what corresponds to n, c, and k here. I think you need to come up with a better way of standardizing metrics, since saying that you compare pass@1 of your system with pass@4 rate of the baseline is quite confusing. The reason I am trying to figure this out is because in general making a claim of a good pass@1 rate is a really strong claim.

---

> > > > > ### Author Response · Authors · 2024-11-23
> > > > >
> > > > > We have updated the PDF, yes. We can share a version with updated changes once we incorporate the final requests (more analysis) but the evaluation section is basically entirely updated.
> > > > >
> > > > > We use that exact implementation to compute the pass@k metric given any n, c and k. But: this metric is only fair if approaches have the same budget. So: we put the more expensive "iterative" approaches (self-debug, within-prompt search) at a small (aligned) disadvantage compared to the "simple" baselines (straight, chain-of-thought).
> > > > >
> > > > > (In the following examples, we compute aligned pass@4. )
> > > > >
> > > > > When directly generating code, we get 32 completions (n = 32), count how many are correct (c) and then set k = 4 to compute the pass@4 rate (in the main experiment). That answers "if we can get the model to generate 4 programs, what's the probability that one is correct?"
> > > > >
> > > > > In chain-of-thought, we count the "thought" as one additional program. So, we get 16 completions (n = 16), count how many are correct (c) and then set k = 2 to compute the pass@2 rate. That answers "if we can get the model to generate 2 thoughts and 2 programs, for a budget of 4 generations in total, what is the probability that one of the programs is correct?"
> > > > >
> > > > > In within-prompt search, we get 4 lines of code at each iteration. In total, we thus generate 4 "whole programs" after the search completes. It would be unfair to compare our approach in pass@1 to an approach that just generates a single program. So, we repeat our approach for 4 iterations (n = 4), count how many are correct (c) and then compute the pass@1 rate (k = 1). Again, that answers "if we can get the model to generate 4 programs in total, what's the probability that one is correct?"
> > > > >
> > > > > Good pass@1 rates are indeed expected to be hard, but they still can be gamed by using a more expensive method. For example, if I present a method that "samples 128 programs from a model, executes them, and only returns the best one" then it will get a very high pass@1 rate, because all the other completions are "hidden" from the final metric computation.

---

> > > > > > ### Comment · Reviewer_g8hP · 2024-11-27
> > > > > >
> > > > > > Thank you for your clarifications. I am increasing my score.

---

### Official Review · Reviewer_vi7p · 2024-11-03

**Soundness:** 3
**Presentation:** 2
**Contribution:** 2
**Rating:** 6
**Confidence:** 3

**Summary:**

The paper proposes a framework to solve PBE by searching for programs by generating lines of code in straight-line program style. When generating the next lines, they consider the partial program generated so far and its corresponding execution state to generate the next line. The LLM here acts as a policy to generate the next action, i.e., line of operation. They show that LLMs with this framework can increase both solving rate and consistency of solving.

**Strengths:**

* The paper demonstrates that the search and execution improve the performance on different datasets across lists, strings, and basic python programming
* The paper's framework is clean and can be generalized to other programming languages or domains where execution state can be compactly represented

**Weaknesses:**

* Issues with the baselines and comparison
  * There is a simple LLM PBE baseline: search by sampling entire programs and then executing to find the right program satisfying the input-output assertions. For the baseline LLM results ("Free" in the figure), they did not do multiple samples and execution. Because the paper proposes to use line-by-line search/generation and execution to find the program satisfying the input-output, it would be better to know how it compares against a simple entire program resample and execute approach without the proposed line-based search.
  * Other neural or symbolic baselines not presented: they use PBE benchmarks from the symbolic methods literature (flashfill++, playgol, list) but their performance is not listed, so readers would not know if using LLM as a search policy is actually better than traditional methods doing program search.
* The paper only uses one model to perform experiments (GPT-4). It would be nice to add results for other models, especially open models or smaller ones, to see if open models can reproduce the results and whether the model size/capability would affect the effectiveness of the approach.

**Questions:**

- It is not clear how the LLM determines which states (partial programs) to expand. Could you lay out the algorithm of the propose framework so it can be less ambigous? Are all the partial programs being expanded with k actions in the default policy setting?
- "Without execution, the model backtracks less, as it does not realize that the current operations $o_{<i}$ are dead ends, and as such performs worse (see Figure 2)." Can you clarify the mechanism of backtracking? Does backtracking only happen when pruning based on the most recent `k` operations? How is the pruning parameter `k` set?
- How do you ensure that there are no statements with side effects modifying the states, e.g., v3 = (v2=1) then v2 is modified?

---

> ### Author Response · Authors · 2024-11-23
> **Response to review**
>
> Thank you for your review.
>
> Regarding your comment: "There is a simple LLM PBE baseline: [...]"
> - Note that our claim is not to perform better than any domain-specific approach.  Our claim is that (1) language models perform better when they can reason about the intermediate results, and (2) language models can perform within prompt search. These claims result in a method that can be applied to any domain.
> - As outlined in the common response, we will extend the “simple” baseline to a generate-and-test strategy with the same budget as our approach. We will also add results on more generic baselines—such as chain-of-thought, self-debugging and tree-of-thought (out-of-prompt search). See the general comment for more information about additional experiments.
>
> Regarding your comment: "The paper only uses one model to perform experiments (GPT-4). [...]"
> - Empirical, small-scale experiments show that the approach does not work for smaller (weaker) models, because they are not as capable at following the straight-line instruction. We will highlight this limitation in the paper. Fine-tuning a small model as a better policy on synthetic data [2] remains future work, and we will add it as such.
> - [2] [2406.08316] Is Programming by Example solved by LLMs?
>
> Regarding your question: "It is not clear how the LLM determines which states (partial programs) to expand. [...] Are all the partial programs being expanded with k actions in the default policy setting?"
> - All partial programs are simply represented by one big "bloated" program. For example, in Figure 1, the three operations left after de-duplication and canonicalization are simply concatenated to the previous program to obtain the prompt for next iteration.The model can then choose which variables to use, which corresponds to choosing which nodes to expand. In fact, the model can choose not to use any of the newly added v3, v4, v5 variables, which will correspond to (implicit) backtracking. We will add a full example (that builds on Figure 1) to the end of each section.
>
> Regarding your comment: "Can you clarify the mechanism of backtracking? [...] How is the pruning parameter k set?"
> - Backtracking is implicit: we did not use any explicit pruning. A backtrack is simply discarding all previous operations and restarting with the arguments. For example, generating “v9 = a + b” is considered a backtrack, because it is ignoring all values v1 – v8 and “starting from scratch.” We will add a concrete example of backtracking to the paper.
>
> Regarding your question: "How do you ensure that there are no statements with side effects modifying the states, e.g., v3 = (v2=1) then v2 is modified?"
> - We do not have such a check in place. We extract the values from the final state, so any side-effects should be captured.

---

> > ### Comment · Reviewer_vi7p · 2024-11-26
> >
> > Thank you for your detailed response and the revised pdf incorporating more examples and experiments. I'm curious if the aligning part of the k of different methods' pass@k can be done differently. Is it possible to get the average token usage of each method (including or excluding the prompt tokens respectively) and use that to align?

---

> ### Author Response · Authors · 2024-11-26
>
> We have considered different methods, but there are some challenges associated with each of them.
>
> - Input tokens are cheaper and affected by caching. WPS allows a lot of caching (the prompt strictly grows) and is sparing on the output, ToT allows the same caching on the program prompt but no caching on the value prompt, and self-debug allows the least caching.
> - To compute aligned pass@k rates between methods, the ratio between their token usage has to be a whole number, or we need to compute enough iterations to approximate their LCM (i.e., if approach A uses 2.5 as many tokens as approach B, we'd have to compare pass@2 versus pass@5, which requires many iterations to complete).
>
> Instead of aligning pass@k rate, an alternative is to look at pass@1 rate versus token usage. In the following image, this is plotted for WPS and ToT, which reinforces our finding that WPS is token-efficient and ToT scales better for a bigger budget: https://i.imgur.com/1NsSIXL.png (different data points for same approach and dataset correspond to k = 2, 4 and 8). *But:* this comparison is not very informative between search and non-search methods, as the latter have lower pass@1 rates for much lower token usage.
>
> We therefore proposed to introduce the aligned pass@k as a metric that can be used across approaches, trading off a slight imbalance in token use for ease-of-use across experiments and being budget friendly (we can scale to the "most expensive" approach).
>
> The output tokens are compared between WPS and ToT in Figure 5, showing that the aligned metric between them seems fair (potentially slightly favoring ToT). In Figure 4, we use the pass@1 rate to indicate that the value function for ToT scales better for k. This figure contains roughly the same information as the above image, but then in function of a concrete hyperparameter.

---

> > ### Comment · Reviewer_vi7p · 2024-12-03
> >
> > I increased the score based on the update. However, there are still some main concerns:
> >
> > ## Baselines May Be Too Weak
> > The baselines considered in the paper may be too weak. For example, for the SyGuS benchmark, the method in this paper scored about 75%, but in the Hypothesis Search[1] paper which the paper has referenced, it's 94% and not using too many samples. Quote from [1]:
> > ```
> > We find that GPT-4 can generate correct programs for 94.3% of the SyGuS tasks using 8 programs with two rounds of feedback without hypothesis generation, demonstrating strong performance in a direct program generation approach
> > ```
> > ## Only Using GPT-4o
> > Although the author talks about this in the limitation section, I think it is important to show that the method can be generalized to other LLMs, especially now that other LLMs with similar coding capabilities are available via API or open models.
> > It would be better to show whether it works on other LLMs or not to rule out specific post-training techniques unique to GPT-4o.
> >
> > [1] Hypothesis Search: Inductive Reasoning with Language Models, Wang et al, ICLR 2024

---

> > > ### Author Response · Authors · 2024-12-03
> > >
> > > Thank you for continuing the discussion.
> > >
> > > ## Baselines
> > >
> > > The performance gap on SyGuS can be explained by three factors: benchmark subset, metric, and controlled experiments.
> > > 1. We start from a more complete set of 205 benchmarks across different iterations of the SyGuS competition and filter them down to 94 non-trivial cases (by sampling 5 programs at t=0.6 with a single given example and pruning cases with a pass@1 of 1.0). This allows us to perform more experiments (saving cost and time) on more challenging cases.
> > > 2. Eight programs and two rounds of feedback requires 8 * 3 = 24 generations, and we could perform 6 rounds of WPS with n = 4 with the same budget. We would therefore need the success or failure of each program to compare numbers, further motivating the need for (token-)aligned metrics in PBE with LLMs.
> > > 3. In our paper, we report controlled experiments on straight-line code generation to demonstrate the effect of execution and search. On our SyGuS subset, the aligned pass@8 rate for the self-debug strategy (~ iterative process from the Hypothesis Search paper) of free-form code generation of 88.7% goes down to 78% for straight-line code generation. Full results of this comparison across three strategies can be seen in this figure: https://i.imgur.com/wO9G2lX.png. This motivates our extension of WPS to free-form code generation in future work (and shows that syntax plays a role in the model's ability to reason about code). Interestingly, for Lists, straight-form code generation works better because it removes the "looping" bias of the model. We are happy to add this figure to related work to offer full transparency.
> > >
> > > Note that we also implemented Hypothesis Search [1] and allow the use of the hypotheses in any of the proposed methods. During preliminary experiments on our benchmarks, similar to our findings, the hypotheses only performed better on the Lists benchmark on aligned (pass@16) metrics when compared to other methods. Because it requires a "search" per hypothesis, performing large-scale experiments is prohibitively expensive (especially if we want to generate at least a few hypotheses).
> > >
> > > ## Models
> > >
> > > We agree that extending the evaluation to multiple models would benefit our evaluation. Unfortunately, we do not have the resources to perform a large-scale evaluation (all baselines and all folds) on different models across vendors. Note that Hypothesis Search was also evaluated on GPT-4 only. Similar to their work, we can carry out a relevant subset of experiments on gpt-4-turbo and add them to an appendix.

---

### Official Review · Reviewer_MDTt · 2024-11-03

**Soundness:** 3
**Presentation:** 3
**Contribution:** 2
**Rating:** 6
**Confidence:** 4

**Summary:**

The paper presents “within-prompt search” method to enable LLMs to reason about the syntax and semantics of programs concurrently. This approach is a novel instantiation of PBE with execution guidance specifically designed for LLM-based program synthesis. The within-prompt search technique encodes multiple candidate code completions within a single prompt. This enhances the likelihood of producing syntactically and semantically accurate code.

The authors position the role of the LLM as a policy to generate code and as a value function to evaluate previously generated code based on its semantics (concepts from reinforcement learning theory where the policy guides generation and the value function assesses the quality of partial solutions). The value of these partial programs is determined simply by syntactic validity and consistency with given examples.

Two baselines are used for comparison: an unrestricted baseline without any search constraints or execution annotations and a second baseline where straight-line constraints are imposed, but search and annotations are disabled. The experimental results show a significant improvement in accuracy and consistency with the within-prompt search method.

**Strengths:**

1. The proposed method is straightforward, yet the experimental results demonstrate substantial improvements over the baselines.
2. The paper is well-written and easy to follow.

**Weaknesses:**

1. Since partial programs are imperative, the semantic annotations include the results of executing each line of code on example inputs, which can grow combinatorially. The paper does not address the limitations of this approach, particularly given the limited window size in LLM prompts and the increased risk of hallucination as prompt size grows.
2. The baselines, which use unrestricted and straight-line constraints without search or annotations, may not fully highlight the method's advantages. Incorporating other PBE or LLM-based synthesis techniques with integrated search mechanisms could be helpful.
3. The paper's within-prompt search mechanism utilizes an implicit backtracking method, but it’s unclear how effectively this avoids redundant or cyclic paths, which could degrade efficiency.

**Questions:**

1. Does this approach work only for imperative programming? It seems it might be more suited for functional or recursive code structures.
2. In Line 244, you mention, “We remove any duplicate lines of code based on their execution results.” Does this apply even if lines are syntactically or semantically different beyond the given examples? Could this lead to false negatives in your experiments?
3. Does your approach inherently favor simpler solutions, or can it dynamically adjust priorities based on feedback? Can you do an analysis on the complexity of the synthesized programs?
4. Since the paper uses standard PBE benchmarks, it would be beneficial to include comparative results from other approaches (e.g., traditional program synthesis methods) on these benchmarks.
5. How does the method handle dead ends—situations where intermediate results do not align with any possible valid completion? A deeper discussion on techniques like pruning unpromising paths or resetting to prior valid states would improve your paper.
6. What do the confidence belts represent in Figure 5?

---

> ### Author Response · Authors · 2024-11-23
> **Response to review**
>
> Thank you for the review. Here we answer the specific questions raised in the review (beyond what we have discussed in the common response above.)
>
> Regarding your comment that semantic annotations can grow exponentially, you are correct that even though the length of the program only grows linearly in terms of the number of iterations, there is a “risk” of intermediate execution results growing very large. This is not very common and has not happened in our experiments. For completeness, we propose to perform an analysis of the prompt size: (1) the number of tokens consumed by code and comments and (2) the probability of generating a correct line as a function of the size of the prompt.
>
> Regarding your comment: "The baselines, [...], may not fully highlight the method's advantages. Incorporating other PBE or LLM-based synthesis techniques with integrated search mechanisms could be helpful."
> As mentioned in the common response, we will add the following outside-prompt search baselines:
> - Tree-of-thought, which corresponds to your suggestion of the model explicitly picking an operation at each iteration (explicit value function).
> - Self-debug loop, where we use the conversational paradigm of language models to provide execution results (or exceptions) until a correct program is found.
> Note that a key feature of within-prompt search is reducing the number of tokens required to perform the search, so we will consider a similar budget for all baselines. See the general comment for more information.  Comparing with domain-specific PBE solvers is not in scope as it is orthogonal to the point of the paper.
>
> Regarding your comment: "The paper's within-prompt search mechanism utilizes an implicit backtracking method, but it’s unclear how effectively this avoids redundant or cyclic paths, which could degrade efficiency. "
> - Cyclic paths are avoided by pruning semantically equivalent operations: if v6 has the same outputs as v1, then v6 is removed. We don’t explicitly handle redundant paths, but we assume that the model will simply ignore nodes that it thinks are not useful.
>
> Regarding your question: Does this approach work only for imperative programming? It seems it might be more suited for functional or recursive code structures.
> - The main requirement is being able to execute partial code, so it requires a “bottom-up” paradigm. An extension can be made to functional programming, where we ask the model to generate and combine functions and provide intermediate outputs on the function level—we leave this as future work. Recursive structures only execute once a base case is reached, which does not allow intermediate results to be shown.
> Whereas the scope of programming style has some constraints, there are other applications of within-prompt search with execution, for example, program repair, auto-completion or code generation from only natural language and inputs. The “challenge” is replacing the soundness check during the search with another stopping criterion, for example, asking the model to output a [STOP] token. We will add this to future work.
>
> Regarding the question: In Line 244, you mention, “We remove any duplicate lines of code based on their execution results.” Does this apply even if lines are syntactically or semantically different beyond the given examples? Could this lead to false negatives in your experiments?
> - We only use the examples given. Technically, it can thus happen that lines that are not identical (on inputs outside of the given examples) are removed, which can negatively affect performance. One way to guard against this is to use execution based on the examples and any additional inputs that are available. In this paper, like previous work, we assume that all examples are given [1].
> - [1] [2309.05660] Hypothesis Search: Inductive Reasoning with Language Models
>
> Regarding the question: "Does your approach inherently favor simpler solutions, or can it dynamically adjust priorities based on feedback? Can you do an analysis on the complexity of the synthesized programs? "
> - This is an interesting question. Because we check each operation for soundness on the given examples, it is arguably more “breadth first” than generating a straight-line program. We will compare the (canonical) program length of the “Straight” and “WPS” approach to see if we find simpler programs.

---

> > ### Author Response · Authors · 2024-11-23
> > **Continuation of response**
> >
> > Regarding your question: "Since the paper uses standard PBE benchmarks, it would be beneficial to include comparative results from other approaches (e.g., traditional program synthesis methods) on these benchmarks. "
> > - Note that our claim is not to perform better than any existing or domain-specific approach.  Our claim is that (1) language models perform better when they can reason about the intermediate results, and (2) language models can perform within prompt search. These claims result in a method that can be applied to any domain.
> > - We have added controlled experiments on other execution-guided (self-debug) and search-based (tree-of-thought) approaches, as well as a chain-of-thought baseline. See the general comment for more information about additional experiments.
> >
> > Regarding your question: "How does the method handle dead ends—[...]? A deeper discussion on techniques like pruning unpromising paths or resetting to prior valid states would improve your paper."
> > - It is hard to distinguish between a dead end and an intermediate result, which is why we rely on the model to act as a value function. We have performed an experiment that forces 2, 4 or 8 unique operations in each iteration, which will cause more dead ends. We observe that an explicit value function and outside-prompt search, which removes potential dead ends, performs better as the number of operations in each iteration increases. When the problem statement is less obvious, like in list problems, the ability to try out different things simultaneously and simply ignoring dead ends works better.
> > - In our evaluation, we observed no concerns related to prompt length. We note that we have the option of pruning if prompt length becomes a concern, but since it is tangential to our main contribution, we leave this study for future work.
> >
> > Regarding your question: "What do the confidence belts represent in Figure 5?"
> > - They are 95% confidence intervals around the regression lines. We will make this explicit.

---

> > > ### Comment · Reviewer_MDTt · 2024-11-25
> > >
> > > Thanks for your response and clarifications. I do not have any further questions at this moment.

---

> > > > ### Author Response · Authors · 2024-11-27
> > > >
> > > > With respect to your original question about complexity, we have analyzed the complexity of the code generated using within-prompt search (k = 4) and straight-line generation without search.
> > > >
> > > > Here is a comparison of the distribution of AST depths of the un-straightened code (`v1 = a+b; v2 = v1 + 2` -> `(a+b) + 2`): https://i.imgur.com/u2vteKa.png. There does not seem to be a significant difference, with direct generation yielding *slightly* simpler programs.
> > > >
> > > > The effect is slightly larger when counting lines of code: https://i.imgur.com/IWgYjSE.png. We hypothesize that the short instruction added to use variables as a scratchpad to store intermediate values and add comments with their results bias the model towards generating shorter lines of code to inspect their input.

---

### Official Review · Reviewer_7p18 · 2024-11-04

**Soundness:** 3
**Presentation:** 2
**Contribution:** 3
**Rating:** 6
**Confidence:** 3

**Summary:**

This paper builds on past work on program synthesis, in particular at the intersection of language model guided synthesis and execution guided synthesis. The authors propose a synthesis technique called *execution-guided within-prompt* search. It is execution-guided in that after each line of Python code is written by the language model, it is executed and the result is appended as a comment to the line. And the *within-prompt* technique is based on the idea of taking several single-line completions from the model, concatenating them into a sequence of lines (despite being completions for a single line – of course renaming is done to avoid conflicts), so that at the next step of search the language model will implicitly pick which one (or more) values to use downstream. They evaluate the model and ablations on a range of datasets and find considerable benefits.

**Strengths:**

- The core ideas are interesting and useful. The idea of appending execution information makes sense, and is very related to past work (which the authors cite) on guiding synthesis using execution results of program prefixes – it's surprisingly not an idea that I have seen used with LLMs in particular, where the execution information can be shown as a string for many data types, but it makes a lot of sense to do it and is found to be beneficial here.
    - Some more related work: ExeDec (Shi et al 2024) is a relevant piece of related work – they don't use the execution of the program prefix to guide execution, rather they predict the execution result of the next step ("subgoal") they want to accomplish in the program and then write code conditioned to try to achieve that. Generally work on code repair is also very related – while it involves writing whole functions at once and seeing their execution results and error messages, it's still quite related.
- The within-prompt search idea is also interesting – and it's intriguing that it sometimes uses more than one intermediate value produced in a single step downstream. It's also nice how the model can recover when it messes up by implicitly backtracking just by building on some earlier execution result.
    - It wasn't obvious to me that this search strategy would work well, given that this is not at all a style of coding that I would expect to see much of in internet training data (often *all* intermediate variables are used, developers don't write extra unused variables too often) – so it's nice to see that it works.
- Figure 4 is nice – it's helpful to see how execution leads to more backtracking, presumably because the model has seen its mistake through the execution result and takes a different approach
- A quite wide range of benchmarks are used, a solid set of benchmarks for a synthesis paper

**Weaknesses:**

- The within-prompt search idea is interesting, but it'd be nice to see it compared to other methods for performing the search, where the execution information is still included. One very simple baseline that's not even a search would be to have the LLM explicitly pick which completion to keep, at which point the others are discarded from the context, and the program is written in this new context (as opposed to having all the other choices in the context still). That would produce rollouts instead of a search, but is still a good baseline to have. There is other work, like Tree Of Thoughts (Yao et al 2023) and Stream of Search (Gandhi et al 2024) that have focused on using LLMs for tree search. Comparisons to other methods, whether it's those ones or other tree-search baselines, would strengthen the evaluation. Even if just relatively simple LLM-search-based baselines were used, it would help with making it clear what sorts of alternatives exist to within-prompt search and how it measures up to them (or why none of them apply – but I expect some other simple form of search would apply). In short, it'd be helpful to have experiments that compare within-prompt search to outside-prompt search, to clarify the advantages of this contribution.

> ...if the policy is certain and samples with less diversity, the model is more likely to solve a problem in the end. Conversely, if the policy samples many diverse lines, it is less likely to solve the problem in the end. This indicates that the model does behave like a policy.
- While I agree that the LLM is certainly a policy (it's sampling the next action), this explanation confused me. Correlation between diversity and success rate doesn't mean something behaves "like a policy" – you could certainly have a policy that doesn't have this behavior, and through temperature you can modulate how much diversity you have in a policy, but that's separate from whether it behaves "like a policy". Maybe the authors meant it behaves like a certain kind of policy?

> ...keeps on sampling until we have k = 4 syntactically unique operations. Having more possibilities requires the model to act as a better value function. For 4 out of 5 datasets, the performance slightly increases, reinforcing our intuition that the model correctly picks which option to continue from
- I also agree that the model is implicitly acting as a value function, as it picks what to extend, which is interesting. But I don't follow the statement that having more possibilities "requires" acting as a "better" value function. Having more low-quality options requires a better value function to weed out the bad ones, but having more from the same distribution shouldn't have that effect. Having more of the same quality options *does* require a good value function in order to *benefit* from the extra examples – and this might be what the authors meant, and the wording just confused me.
  - That said, the improvements in Figure 6a are fairly small. As a thought on an experiment that might show larger differences, it'd be interesting to see whether something like only giving it 2 unique options and jumping up to 4 or 6 unique options would have a bigger improvement (and not focusing on the unique vs nonunique distinction).

- I think that Figures 2a and 2b would likely be better as bar plots instead of a line plots, since the X axis is categorical data not a continuous metric.
- Figure 3 might be better as a heatmap since it's discrete data so arbitrary points aren't needed, and it would help the problem where the density of points is so high that many just look like solid red bars and it's hard to compare them.

- The approach is limited to working with straightline code – but could perhaps be extended to write whole units of code at once containing if statements or loops as long as they're all done in one completion, and there are other interesting directions to build off of there
- The statements about soundness guarantees in the abstract are confusing – and soundness is mentioned far more in the abstract than the paper. The soundness guarantees come from executing the programs to check correctness, which is done when testing programs generated by LLMs, whether or not they were constructed iteratively through the suggested approach here or all at once – that doesn't change the soundness in my understanding.
- line 31: "grail" -> "holy grail"
 > We leverage pre-trained language models to perform programming-by-example without restrictions on the domain (specific language).
- This first line of the contribution list is not itself a contribution of the paper to the field since it's shared by other LLMs synthesis work – it's a feature of the second line (execution-guided within-prompt search)

**Questions:**

- Is the "8 iteration" limit used for the "Free" and "Straight" settings too? In particular I want to make sure they're not also restricted to only producing 8 lines, which would limit them compared to the "search" based models that are allowed to use more than one line from a single iteration in their solution.

---

> ### Author Response · Authors · 2024-11-23
> **Response to review**
>
> We truly appreciate your suggestion for comparing within-prompt search with outside-prompt search. As mentioned in the common response above, and as can be seen in the revised PDF, we add the following outside-prompt search baselines:
> - Tree-of-thought, which corresponds to your suggestion of the model explicitly picking an operation at each iteration (explicit value function).
> - Counter-example guided loop, where we use the conversational paradigm of language models to provide execution results (or exceptions) until a correct program is found (up to some number of iterations). We additionally sample N programs at each iteration and pick the one with the fewest errors for the next round (roughly turning the search into a beam search of code repair).
>
> The new evaluation shows that our "within-prompt search" is the best choice when we give a similar output-token budget for all baselines. Tree-of-thought can perform better, but only when it is given a higher output-token budget. Please see the general comment and revised PDF for more information.
>
> Regarding your comment: "While I agree that the LLM is certainly a policy (it's sampling the next action), this explanation confused me. [...] Maybe the authors meant it behaves like a certain kind of policy?"
> - We agree here. Technically any function that returns an operation is a policy. We mean that it behaves like a “trained” policy that actually prefers operations that are more likely to have a high value. We will make this explicit.
>
> Regarding your comment: "I also agree that the model is implicitly acting as a value function, [...] As a thought on an experiment that might show larger differences, it'd be interesting to see whether something like only giving it 2 unique options and jumping up to 4 or 6 unique options would have a bigger improvement (and not focusing on the unique vs nonunique distinction). "
> - Because the model continues sampling until it finds 4 unique operations, this shifts the distribution to having more low-quality operations (as a kind of rejection sampling). We have carried out an experiment for 2, 4 and 8 operations, both in a diverse and non-diverse settings, and show that (1) the implicit value function remains effective, but (2) an explicit value function (ToT) scales better for higher k.
>
> Regarding your comment: "I think that Figures 2a and 2b would likely be better as bar plots instead of a line plots, since the X axis is categorical data not a continuous metric. "
> - We have added separate figures for comparison with baselines (new figure) and ablations (current figure 2a and 2b) and change them into bar-plots.
>
> Regarding your comment: "Figure 3 might be better as a heatmap since it's discrete data so arbitrary points aren't needed, and it would help the problem where the density of points is so high that many just look like solid red bars and it's hard to compare them."
> - We have updated to a heatmap-style plot.
>
> Regarding your comment: "The approach is limited to working with straightline code – but could perhaps be extended [...]"
> - This is indeed an interesting direction that we will leave for future work (and explicitly mention as future work). The challenges to be solved are how to represent the intermediate values for control flow statements. But we note here that straight-line code is very expressive in Python, with functions like `itertools.count` even supporting infinite loops (we allow functions from the standard library) and the “x if condition else y” construct supporting conditions.
>
> Regarding your comment: "The statements about soundness guarantees in the abstract are confusing – [...] – that doesn't change the soundness in my understanding."
> - That is correct. We meant (and have clarified) that building a program line-by-line with execution results is a more efficient way of obtaining a sound program than rejection sampling.
>
> Regarding your comment: "This first line of the contribution list is not itself a contribution of the paper to the field since it's shared by other LLMs synthesis work – it's a feature of the second line (execution-guided within-prompt search) "
> - This is a good point; we have updated the contributions accordingly.
>
> Questions
>
> Regarding your question: "Is the "8 iteration" limit used for the "Free" and "Straight" settings too? In particular I want to make sure they're not also restricted to only producing 8 lines, which would limit them compared to the "search" based models that are allowed to use more than one line from a single iteration in their solution."
> - No, in free and straight the model can just generate one large program without any limitations on program length (max tokens is set to 2048). We have removed free-form generation from the controlled experiments since it was difficult to account for token budgets there. We will added separate results on free-form code generation and use it to motivate more future work.

---

> > ### Comment · Reviewer_7p18 · 2024-11-25
> >
> > Thank you for the thorough responses! You refer a few times to the revised PDF, but I don't actually see a revised PDF (with heatmaps, tree of thought baseline, etc). I believe I'm looking in the right place, because on other submissions I'm reviewing I see the revised version.

---

> > > ### Author Response · Authors · 2024-11-25
> > >
> > > We had mistakenly re-uploaded the previous version, thank you for bringing this to our attention. This has now been corrected with a new version. Note that we are still in the process of continuing the motivating example across Section 3, as well as adding additional insights (complexity of code versus straight code generation without execution feedback, number of tokens used by the input).

---

> > > > ### Comment · Reviewer_7p18 · 2024-12-03
> > > >
> > > > Thank you for all the responses and considerable revisions to the evaluation and addition of several important baselines, I've raised my score.

---

### Author Response · Authors · 2024-12-02

We would like to remind the reviewers of the updated PDF, which addresses the concerns around evaluation and clarity. Because we did not explicitly mark changes in the PDF, here is a high-level overview.

- The contributions are centered around within-prompt search and the evaluation.
- We added a leading example in four parts throughout the approach (sections 3.1 - 3.4), which shows how a single prompt is updated with different candidate programs and their execution semantics.
- We extensively reworked the evaluation (section 4.2 and 4.3) based on the suggestion of comparing with outside-prompt search, which neatly corresponds to tree-of-thought. We propose the *aligned pass@k* metric as a way to make a fair comparison based on output token usage, based on the reviewers' suggestion that simple code generation should be allowed to perform a generate-and-test strategy.
- We include limitations and future work (section 6) that (1) explicitly states the dependency on powerful models that adhere to straight-line instructions, (2) provides insights on how to scale within-prompt search to free-form code generation, and (3) highlights that within-prompt search can be applied to program repair and auto-completion in environments where inputs are available.

---

### Meta-Review · Area_Chair_NRnW · 2024-12-19

**Metareview:**

The paper deeply investigates a way of showing intermediate execution results to language models in order to improve their ability to generate code from examples across an unusually broad range of challenging programming by examples problems, including hard real world benchmarks like FlashFill++. One primary strength of the paper is the breadth of its evaluation, and algorithmic contributions to the structure of execution guided synthesis, in particular: creating a "bloated" program that contains multiple past generations for a neural network to pick between, and potentially even recombine, allowing some of the combinatorial search to be offloaded to the LLM's decision-making. The main weaknesses include a narrow conceptual contribution (essentially: the method is prompting+execution guidance, albeit with the previously mentioned cleverness in "bloating" the program). The method itself yields narrow improvements, which is not ideal but not disqualifying.

I recommend accepting the paper despite those weaknesses: It contains conceptual novelty, and can encourage the community to consider a broader range of benchmarks. In its current form however the contribution remains narrow, but would be significantly enhanced by showing how to do within-prompt execution guidance for programs with control flow, or how to apply it to code repair: these are suggested currently in future work, but they would have significantly increased the conceptual and technical contribution to include even a modest demo within the present submission.

**Additional Comments On Reviewer Discussion:**

The discussion was productive: it resulted in new revisions for clarification, and new analyses. I weigh the new analysis which compares for token cost quite heavily, as this is fundamentally a search problem and the claim is fundamentally about compute efficiency.

---

### Decision · Program_Chairs · 2025-01-22

Accept (Poster)